# Genetic Profiling of Idiopathic Antenatal Intracranial Haemorrhage: What We Know?

**DOI:** 10.3390/genes12040573

**Published:** 2021-04-15

**Authors:** Anna Franca Cavaliere, Irene Turrini, Marta Pallottini, Annalisa Vidiri, Laura Marchi, Federica Perelli, Simona Zaami, Giovanni Scambia, Fabrizio Signore

**Affiliations:** 1Department of Obstetrics and Gynaecology, Santo Stefano Hospital, Prato, 59100 USL Toscana Centro, Italy; annafranca.cavaliere@uslcentro.toscana.it (A.F.C.); irene.turrini@uslcentro.toscana.it (I.T.); laura.marchi@uslcentro.toscana.it (L.M.); 2Department of Woman and Child Health, Fondazione Policlinico Universitario A. Gemelli IRCCS, 00168 Rome, Italy; annalisavidiri@gmail.com (A.V.); giovanni.scambia@policlinicogemelli.it (G.S.); 3Department of Obstetrics and Gynaecology, Santa Maria Annunziata Hospital, Florence, 50012 USL Toscana Centro, Italy; federica.perelli@uslcentro.toscana.it; 4Department of Anatomical, Histological, Forensic and Orthopedic Sciences, “Sapienza” University of Rome, 00100 Rome, Italy; simona.zaami@uniroma1.it; 5Department of Obstetrics and Gynaecology, Santo Eugenio Hospital, ASL Roma 2, 00100 Rome, Italy; fabrizio.signore@aslroma2.it

**Keywords:** antenatal intracranial hemorrhage, intraventricular hemorrhage, hemostatic genes, collagen genes, X-linked-GATA1 genes

## Abstract

Intracranial hemorrhage (ICH) is reported in premature infants and rarely, in prenatal life. Fetal ICH can be accurately identified in utero and categorized by antenatal sonography and/or MRI. Infectious disease, maternal drug exposure, alloimmune thrombocytopenia, maternal trauma, coagulation disorders and twin-to-twin transfusion syndrome can cause fetal ICH. However, in many cases, the cause is not identified and a genetic disorder should be taken into consideration. We conducted a review of the literature to investigate what we know about genetic origins of fetal ICH. We conducted targeted research on the databases PubMed and EMBASE, ranging from 1980 to 2020. We found 311 studies and 290 articles were excluded because they did not meet the inclusion criteria, and finally, 21 articles were considered relevant for this review. Hemostatic, protrombotic, collagen and X-linked GATA 1 genes were reported in the literature as causes of fetal ICH. In cases of ICH classified as idiopathic, possible underlying genetic causes should be accounted for and investigated. The identification of ICH genetic causes can guide the counselling process with respect to the recurrence risk, in addition to producing relevant clinical data to the neonatologist for the optimal management and prompt treatment of the newborn.

## 1. Introduction

Intracranial hemorrhage (ICH) is reported in premature infants but may occur, rarely, in prenatal life as well. The reported incidence of in utero ICH varies from 1/100,000 up to 1/1000 [1].

The classification of ICH includes five types according to the site of the lesion: intraventricular (IVH), subarachnoidal, intraparenchymal, cerebellar and subdural hemorrhage.

IVH is the most common variety of neonatal ICH, due to the characteristics of the immature brain. They are subdivided into four grading, based on the extent of the lesion: the first three grades are limited to the ventricles, while the fourth grade includes parenchymal involvement.

The diagnosis of fetal ICH is usually performed in the third trimester. Considering that the most common variant is IVH, this finding can be explained by embryologic development. In fact, vascular connection between germinal matrix and the subependymal venous network, which are bleeding sites in IVH, is clearly present only after 20 weeks of gestation [2].

Fetal ICH can be accurately identified in utero and categorized by antenatal sonography and/or MRI. Ultrasounds are the modality of choice in the diagnostic pathway of fetal ICH, especially in IVH variants. Based on ultrasonographic findings, in 1996, Vergagni et al. proposed an IVH classification framework and a scoring system [3].

Typical ultrasonographic signs of IVH are various degrees of ventriculomegaly with irregular bulky choroid plexus; hyperechogenic, sometimes indented ventricular walls; intraventricular hyperechogenic foci suggesting clots; a subependymal hemorrhage manifesting as periventricular echodensities; and an intraparenchymal hemorrhage appearing as irregular echogenic brain mass, whereas under normal circumstances non-echogenic images would have been visualized (such as thalami). A subdural hemorrhage is easily recognized due to the presence of an echogenic area representing the hematoma that displaces the Sylvian fissure from the inner table of the skull, and finally, in the case of intracerebellar hemorrhage, the ultrasonographic findings included abnormal echogenicities within the posterior fossa [4].

The value of MRI as a diagnostic tool for fetal ICH is still debated [5].

It has been reported that a small periventricular hemorrhage is better recognized by MRI rather than ultrasounds; additionally, MRI could help in detecting the timing of bleeding and the progression of hematoma.

Among the possible etiologies of fetal ICH, we can list infectious disease, maternal drug exposure, alloimmune thrombocytopenia, maternal trauma, coagulation disorders and twin-to-twin transfusion syndrome. However, in many cases, the cause may not be identified, and a condition arising from a genetic disorder, associated with an increased risk for cerebral arteriopathy, should be taken into consideration.

Genetic causes of fetal ICH include:

-Hemostatic genes: von Willebrand’s disease, congenital factor V, factor VII, factor VIII and factor X deficiency and protrombotic disorders (factor V Leiden, MTHFR mutation, protein C deficiency);-Inflammatory genes: polymorphisms in the pro-inflammatory cytokine IL-6;-Collagen genes: COL4A1 and COL4A2 mutations;-X linked GATA1 gene mutation.

Detecting the cause of the intracranial hemorrhage during pregnancy can offer better neonatal management and evaluate reproductive risk in future pregnancies.

We conducted a review of the literature to investigate what we know about the genetic origins of fetal intracranial hemorrhage.

## 2. Materials and Methods

We performed research on the PubMed and EMBASE databases, ranging from 1980 to 2020. The search terms included the combination of antenatal or fetal or prenatal intracranial hemorrhage with genetic or molecular or gene. We also used Mesh terms in order to fine-tune the search, and evaluated all the references in order to retrieve potentially relevant studies and reports. We used Mendeley to upload literature and select related articles.

## 3. Results

In literature there are limited data on this topic. We came by 311 studies, but out of those, 290 articles were excluded because they included post-natal ICH, unrelated to the genetic causes of ICH or based on experimental models. We included only articles in the English language. Ultimately, 21 articles were deemed suitable for this review.

We collected data on: hemostatic genes (factor V deficiency, von Willebrand’s disease, factor VII deficiency), protrombothic genes (factor V Leiden, MTHFR mutation, protein C deficiency), collagen genes (COL4A1 and COL4A2 mutations), and X linked GATA1 gene mutation. Data obtained are summarized in Table 1.

### 3.1. Haemostatic Genes

Hemostatic factors as causes of ICH have been fairly extensively investigated in children and adults. On the contrary, in the literature there are only a few studies, mainly case reports, dealing with the association between fetal ICH and bleeding disorders.

#### 3.1.1. Factor V Deficiency

Severe FV deficiency (with <1% FV) is a rare disorder with an estimated incidence of one per million. Inheritance is autosomal recessive and consanguinity has been observed in some families of affected patients. Factor V deficiency was first described and labeled parahemophilia by Paul A. Owren in 1947 [6]. Clinical bleeding problems have been reported as less severe than in many hemophiliacs, although intracranial bleeding has been observed. In factor V deficiency clotting occurs slowly. In the literature, we found only two reported cases of ICH caused by FV deficiency.

The first case of severe factor V deficiency in a girl of non-consanguineous Sikh parents with severe antenatal intracranial hemorrhage and subsequent hydrocephalus, between 28 and 30 weeks of gestation was reported by Whitelaw et al. in 1984. Postnatal investigations showed very low (2%) factor V concentrations. The parents were assumed to be heterozygous, due to the mutation with variable penetrance [7].

More recently, Ellestad et al. described a patient who presented an intracranial hemorrhage detected in utero at 36 weeks of gestation, who was ultimately diagnosed with severe factor V deficiency. An extensive intraparenchymal bleeding led to his death, and factor V was undetectable [8].

#### 3.1.2. Von Willebrand’s Disease

Von Willebrand’s disease is a rare, inherited bleeding disorder characterized by defective platelet adhesion and subsequent coagulation defect caused by mutations in the VWF gene. Three main subtypes are described based on the type of von Willebrand factor defect: partial (type 1) or total (type 3) deficiency, and qualitative/functional anomalies (type 2). The inherence is most commonly autosomal dominant but type 3 and in some of type 2 the disease is transmitted by an autosomal recessive manner.

We found in literature only one case of a fetus affected by periventricular hemorrhage (PVH) due to type 2 Von Willebrand’s disease. PVH is characteristic of the immature brain and the early days of extrauterine life, probably due to maladaptation of the cerebral blood flow to extrauterine life. The incidence of PVH in fetal life seems to be fairly low, and in most reported fetal PVH cases, evidence showed a hemostatic disorder. The father was known to have a type 2 vWD (factor VIIIc activity 56%, von Willebrand factor activity less than 5%), while the mother had no evidence of a hemostatic disorder or other abnormalities [9].

#### 3.1.3. Factor VII (FVII) Deficiency

Inherited FVII deficiency is a rare disorder that is inherited as an incompletely recessive autosomal trait; the estimated incidence is 1 in 500,000, with a higher prevalence in populations with a sizeable degree of consanguinity. The clinical manifestations vary widely, and do not always correlate with the degree of deficiency; patients with levels of FVII activity less than 1% usually experience severe bleeding, for which daily life-long infusions with FVII concentrate are often necessary [10].

The only case was reported in 2009 by Landau et al. and it described a consanguineous Bedouin–Arabic couple who experienced the deaths of their two baby girls, one with prenatal hydrocephalus secondary to intracranial hemorrhage and the other with postnatal intracranial bleeding, and both with severe congenital FVII deficiency (less than 1% activity). Genetic analysis revealed that both parents were heterozygous and both daughters homozygous for a point mutation gG9639A in exon 7 [11,12].

### 3.2. Protrombothic Disorders

The role of inherited thrombophilia in the pathogenesis of fetal intracerebral hemorrhage is still controversial. In the last few years, fetal abnormalities in coagulation and fibrinolysis have received increased attention as predisposing factors to antenatally intracranial hemorrhage. Co-inheritance of more than one thrombophilic polymorphism is associated with a greater risk of thrombotic events compared to a single polymorphism inheritance.

Genetic thrombophilia may predispose patients to the onset of cerebral venous thrombosis, after birth at least in preterm infants, and during infancy and childhood. The most common thrombophilic mutations associated with IVH in adults include the factor V Leiden (FV-L) variant, polymorphisms of the methylenetetrahydrofolate reductase (MTHFR) gene and the prothrombin 20210G > A variant.

FV-L mutations have a high prevalence in the general population and the authors concluded that the harmful effects of this thrombophilia may even occur at the earlier stage of fetal development.

Crespin et al. reported an intraventricular and intraparenchymal cerebral hemorrhage associated with cystic leukomalacia in a 24-week fetus with two inherited risk factors of thrombophilia, a heterozygous Leiden mutation of the factor V gene and a heterozygous mutation of the protein C gene. In light of the severe clinical figures on fetal MRI and the poor prognosis, a termination of pregnancy was performed during the 24th gestational week [13].

Antecedents of thrombosis, and genetic risk factors for thrombosis, were found in the family, in both the mother’s and father’s sides. The mother was heterozygous for the factor V Leiden mutation; the father carried type I protein C deficiency.

Ramenghi et al. described a case of germinal matrix-intraventricular hemorrhage (GMH) in a 24-week fetus diagnosed by MRI [14].

A male infant was delivered by Cesarean section in the 37th gestational week, and was found to be heterozygous for two thrombophilic factors, factor V Leiden and MTHFR mutation. The authors suspected that the fetal GMH-IVH may have resulted from a thrombosis affecting cerebral veins. Maternal family history was negative, while the father of the infant was found to be heterozygous for factor V Leiden and homozygous for MTHFR mutations.

Verduet al. in 2005 described another case of heterozygous factor V Leiden in a fetus with hemorrhagic infarction. An ultrasonography performed at 24 weeks showed a peri-intraventricular hemorrhage on the left side, suggestive of hemorrhagic infarction. This image evolved in a porencephalic cyst but remained steady until birth [15]. Coagulation studies showed activated protein C resistance, and the patient was found to be heterozygous for FV-L.

Complete deficiency of protein C is a rare condition; it can be associated with massive thrombosis, often as early as in utero, neonatal purpura fulminans and disseminated intravascular consumption coagulopathy in infants. Over 160 different point mutations in the protein C gene have been identified in recent years and they determine low concentrations of circulating protein C. If not treated, homozygous protein C deficiency is fatal. The diagnosis of this defect in uterus can allow treating it with protein C transfusions or fetal anticoagulant therapy.

Kirkinen et al. (2000) described a pregnancy with fetal homozygous protein C deficiency. MRI examination at 33 weeks of gestation showed ventricumegaly, small hemorrhagic areas near the lateral walls of the second ventricles and intraorbital thrombosis [16]. Emergency Caesarean section was carried out in the 35/36th gestational week due to cardiotocographic anomalies; multiple placental infarcts were also found through histological examination. Eventually, both parents were diagnosed as being heterozygous for type II protein C deficiency.

### 3.3. Collagen Genes 

#### COL4A1 and COL4A2

Collagen, type IV, alpha1 (COL4A1) and alpha2 (COL4A2) are components of basement membranes, including those of the cerebral blood vessels. COL4A1 is expressed in the brain, muscles, kidneys, and eyes. The spectrum of COL4A1-related disorders is well described, and includes: small-vessel brain disease of varying severity, eye defects (retinal arterial tortuosity, Axenfeld–Rieger anomaly, cataract) and some systemic findings (kidney involvement, muscle cramps, cerebral aneurysms, Raynaud phenomenon, cardiac arrhythmia, and hemolytic anemia) [17,18].

COL4A1 mutation carriers have great diversity in the clinical manifestations of the disease within the same family: it can remain asymptomatic or cause devastating disease. Neonates and children may present with porencephaly, intracerebral hemorrhage, or hemiparesis, whereas adults tend to develop intracranial aneurysms or retinal arteriolar tortuosities [19,20,21].

Mutations of COL4A1 have been shown to cause ICH and porencephaly both in mice and humans. In mice, COL4A1 mutants exhibit structural alterations of basement membranes in many tissues including the brain, the eye and the kidney. Mutated mice and human patients show an increased fragility of brain vessels with a higher risk to develop an ICH in case of birth trauma or with the exposition in utero to antithrombotic agents.

De Vries et al. described in 2009 two cases of preterm infants with antenatal intracerebral hemorrhage and established porencephaly due to the presence of a novel G1580R mutation in the COL4A1 gene. Authors postulate that this hemorrhagic event probably occurred antenatally and not peripartum [22,23,24,25,26].

Meuwissen et al. in 2011 reported on novel COL4A1 mutations occurring de novo in four Caucasian patients with extensive prenatal brain destruction at different gestational ages [27].

Colin et al. in 2013 reported two cases of prenatal ICH associated with cataract. They suggested that COL4A1 mutation should be tested for in fetuses with prenatal ICH, especially in the case of lens abnormalities at ultrasound examination [28].

Matsumoto et al. in 2015 described a case of in utero hemorrhagic brain damage leading to schizencephaly; postnatal genetic investigation revealed a COL4A1 mutation [29].

The first case of COL4A1 associated with ICH diagnosed prenatally was reported by Lichtenbelt et al. in 2012. They used fetal DNA isolated from amniotic fluid and found a de-novo pathogenic missense mutation (G1103R) of the COL4A1 gene [30].

Kutuket al. in 2013 performed a whole exome sequencing of 4 newborns diagnosed with fetal ICH, showing no pathological mutations of COL4A1 and COL4A2 [31].

Weng et al. conducted a systematic study of COL4A1 mutations in sporadic ICH. They tested COL4A1 in 96 patients with sporadic ICH, compared them with 145 ICH-free controls and identified two putative mutations in 96 patients with sporadic ICH [32].

Khalid et al. in 2018 reported a case of fetal ICH with a de novo heterozygous mutation in intron 9 of the COL4A1 (α1) gene demonstrating the sequential development of schizencephaly on MRI in utero [33].

In 2020 Maurice et al. tried to establish the prevalence of COL4A1 and COL4A2 gene mutations in a single-center retrospective study conducted on fetuses with ICH specific phenotypes (severe and/or multifocal hemorrhagic and/or hemorrhagic–ischemic cerebral lesions, of different ages and/or associated with schizenchefaly/porenchefaly). Eighteen fetuses met the inclusion criteria and COL4A 1 mutations were spotted in five cases [34].

### 3.4. X Linked GATA1 Gene Mutation 

It is known that the GATA1 gene located in Xp11.23 is involved in the differentiation of blood cells, and mutations of this gene are linked to thrombocytopenia and dyserythropoietic anemia.

In 2011, Bouchghoul et al. described a GATA1 gene mutation responsible for a fetal ICH diagnosed at 36 weeks of pregnancy and two other stillbirth cousins presented with fetal hydrops and congenital hemochromatosis’ phenotype at the 37th and 12th weeks of gestation. Molecular screening revealed the presence of a c.613G>A pathogenic allelic variation in exon 4 of GATA1 gene in those three male fetuses and their mothers [35].

### 3.5. Inflammatory Genes 

Polymorphisms in the pro-inflammatory cytokine IL-6 have been assumed to be potential genetic modifiers for IVH risk, albeit no cases are as yet reported in literature.

## 4. Discussion

The literature regarding fetal ICH focuses primarily on diagnostic sonographic criteria and neurologic outcome.

With regard to the etiology of this disease, when common causes of fetal ICH cannot be identified, clinicians classify ICH as idiopathic. However, several gene mutations have been found to be associated with this condition, although only a few authors have investigated genetic profiling and reported single cases (Table 1). For the time being, 2 cases have been reported about factor V deficiency, 1 case about fetal vW disease, 1 case of factor VII deficiency, 3 cases of FV Leiden mutation (associated to other polymorphisms), 1 case of protein C deficiency and 1 case of GATA 1 gene mutations. Only a few more cases associated with COL4A1 and COL4A2 mutations have been reported in literature.

The first consecutive series of cases of fetal ICH matching with COL4A1 and COL4A2 was published in 2020 by Maurice et al. [34]. Out of 18 severe and/or multifocal hemorrhagic and/or haemorrhagic–ischaemic cerebral lesions, 5 (27.7%) were found to be positive for COL4A1 mutations. The Authors therefore conclude that COL4A1 and COL4A2 gene mutations should be systematically searched for in cases presenting such features.

On the basis of the results obtained by Maurice et al. it is likely that once genetic profiling is systematically investigated, a high incidence of gene mutations may be found, especially in idiopathic conditions. Detecting the cause of ICH could lead to a tailored and prompt treatment (such as coagulation factor infusion) of affected neonates.

From a medico-legal perspective, intracranial hemorrhages, which range from minor to extremely severe, will most likely give rise to litigation and, on average, very substantial compensatory damages if such complications are proven to have arisen from medical negligence. Legal standards have to be applied in order to prove breach of duty and causation, the elements at the very core of medical negligence and malpractice claims. Undoubtedly, fetal intracranial hemorrhages can be challenging to identify. Sonography is the diagnostic modality of choice. However, since ultrasound has low sensitivity for minor hemorrhages, magnetic resonance imaging (MRI) is routinely carried out to confirm the presence of ICH; moreover, MRI can go a long way towards establishing time and evolution of the bleeding. Sonographic evidence of a fetal intracranial hemorrhage hinges upon the timing of the ultrasound related to the incident. If the bleed begins within 24–48 h of the ultrasound, hyperechoic signals without posterior shadowing should be detectable [36]. ICH diagnostic pathways may prove essential in terms of staving off malpractice charges and adverse litigation results. The diagnostic process, it should be kept in mind, is not a “binary relation” always capable of establishing pathological conditions. Most fetal intracranial hemorrhages are detected at routine prenatal sonography. Most cases, often of high grade, are discovered over the third trimester. Postnatal survival rates are considerably high, but so is the risk of adverse neurologic outcome in most neonates [37].

Demonstrating the genetic etiology in neonates born with in utero ICH classified as idiopathic, allows us to acknowledge this event as unpreventable. To identify the genetic cause of ICH can also guide the counselling about the possibility of recurrence risk in the same family.

## 5. Conclusions

When fetal hemorrhage is diagnosed, a prompt inquiry into the genetic profile of disorders related to ICH must be performed, beginning from family history. A stillbirth with intracranial hemorrhage may result from an unrecognized factor deficiency or mutation. There are genetic aspects to take into consideration in case of fetal ICH like demonstrating that this condition was not preventable. In conclusion, it is ethically appropriate to provide parents with a thorough risk assessment as to the prospects of recurrence in future pregnancies [38]. We do not suggest this genetic profiling routinely and we need more consistent data to select a target population in which this analysis could be useful. However, in case other causes are excluded or in case of recurrence in the same family, genetic profiling could be investigated. We need an international collaboration to collect all cases affected by fetal ICH related to these genetic causes in a database.

## Figures and Tables

**Table 1 genes-12-00573-t001:** Genetic causes of fetal intracranial haemorrage: incidence, inheritance and numbers of case reports.

Etiology	Incidence	Inheritance	Case Reports
**Haemostatic genes**			
Factor V deficiency	1/1,000,000	autosomal recessive	2
Von Willebrand’s disease	1/100–1/1000	AD (type 1-2) and AR (type 2-3)	1
Factor VII deficiency	1/300,000–1/500,000	incompletely recessive autosomal	1
**Protrombothic disorders**			
Polymorphism MTHFR gene	3/100–3,7/100	autosomal recessive	1
Factor V Leiden variant	3/100	autosomal dominant	3
Prothrombin 20210G>A variant	3/100–5/100	autosomal dominant	0
Protein C deficiency	2/1000–5/1000	autosomal recessive	2
**Collagen genes**			
COL4A1 and COL4A2	6–7/100,000	autosomal dominant	20
**GATA1 gene mutation**			
	<1/1,000,000	X-linked	1

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
