# Peer review of "Genetic Profiling of Idiopathic Antenatal Intracranial Haemorrhage: What We Know?"

_genes, 2021, doi:10.3390/genes12040573_

Round 1
Reviewer 1 Report
I read with interest the present report, and I found it suitable for pubblication.
However the following criticisms are raised
1) The main criticism is about the final message of the paper. To my opinion focusing attention on the medico-legal implications of an omitted genetic diagnosis is not correct. That is mainly because no consistent nosology can be defined about genotype-phenotype correlations, penetrance, pathomechanisms (monofactorial? oligogenic?) for the majority of the here discussed conditions, with few exceptions. On the other hand the genetic contribution in idiopathic ICH deserves to be investigated. Accordingly, the final message should be about the precise diagnosis and about its scientific relevance.
2) Paragraph on Von Villebrand's disease. Although the pattern of inheritance can be inferred from the text, inheritance and heterogeneous models of inheritance as well, should be specified, for the benefit of readers non highly involved in genetics.
Author Response
Dear reviewer,
Thank you for your suggestions that have improved the quality of our text.
We try to answer all points.
1) We deleted in the text from lines 291 to 298 and from 318 to 328, in order to eliminate the focus on the medico-legal implications of an omitted genetic diagnosis in the discussion and conclusion. We replace “medico-legal aspects” with “genetic aspects” in line 333.
2) Thank you we clarify the concept of inheritance in lines 123-129.
Reviewer 2 Report
This is a review paper attempting to elucidate the incidence and specifics of genetic etiologies (or at least associations) of intracranial hemorrhage. The authors used 21 papers to report examples of ICH that were shown to be associated with genetic findings. They then go on to speculate about the utility of such information in medicolegal proceedings that are seen with neurodevelopmental delays in such cases.
- While a listing of the various genetic etiologies they report is a good start, it does little towards understanding the importance of such by itself. As with any other “test,” in this case US diagnosis of the finding, in order to be useful, it is necessary to put it into the statistical performance metric approach. They report “numerators” of the genetic conditions, they but make no attempt or even speculate as to what the denominators are.
- As part of the above, the paper needs at least 1 summary table of the various etiologies, their incidence, proportion of cases might be attributable to genetics, and how that might affect management.
- Do they recommend that such genetic testing be a “routine” response to ultrasound findings?
- The medicolegal section, while certainly relevant to the modern practice of obstetrics, can’t just hang out there alone. These issues could take up an entire paper or more, and such has been the subject of a lot of literature – for example for cerebral palsy. They should either just state such is an issue well beyond the scope of this paper, or they have to expand considerably which I believe is premature given the points made in #1.
Author Response
Dear reviewer,
Thank you for your suggestions that have improved the quality of our text.
We try to answer all points.
1) We are not sure to have understood the core point of the question. Our review aims to emphasize that there are few case reports of ICH genetic causes nowadays. The limited number of cases in Literature doesn’t permit to elaborate a predictive model of US diagnosis of the findings. In the future this work could represent the starting point to answer to this question by collecting more cases. We add in lines 336-337.
2) We provide this information in table 1. However, we omitted the “management” because we believe that steps outside from the aim of our review.
3) We answer to this question in lines 337-339.
4) We agree with you, we provide to modulate the text in accord to your suggestions as pointed out by the reviewer 1.
Round 2
Reviewer 1 Report
The revised MS can be accepted in its present form.
Author Response
Dear reviewer,
Thank you for your contribution.
Kind regards,
Marta Pallottini
Reviewer 2 Report
WRONG 1. The title genetic “profiling” implies investigation of the underlying molecular changes in the disease. Just listing the Mendelian inheritance pattern is the first column, but there needs to be more. The fact that the authors didn’t even bother to cite the table in the reference is concerning. 2. To use the excuse of there are “scant data” when the title of the paper includes “what we know” is not acceptable. If we don’t know, then state what we do know and what has to be done to find out. To then recommend we don’t do genetic testing because we don’t know what the statistics are will lead to a perpetual ignorance status forever. It is precisely in the cases with no KNOWN genetic syndrome capable of producing intracranial hemorrhage that we most need such genetic testing. 3. Having received now the proper documents that should have been sent, my conclusions are only partially ameliorated. The paper is still disjointed, needs to be restructured, a. Types of anomalies b. What we know about their genetics c. What we don’t know and how that leaves questions that may wind up in the court room d. Developing a database to generate such information.Author Response
Dear reviewer,
Thank you for your interesting suggestions.
Considering that, we performed the following modifications on our paper:
- TITLE: Regarding the title
- we concluded with question mark (?)
- we added the term Idiopathic (to underline that all known causes of Fetal ICH are already been excluded)
- RESULTS: We inserted in the result section the reference regarding our table.
- DISCUSSION: In the discussion :
- we cited the table
- we clarified the medical-legal implications to explain questions that may be underlined in the court room
- CONCLUSIONS: In the conclusions,
- we clarified that this kind of genetic profiling could be investigated when other known causes, including genetic syndromes, are excluded
- we underlined the importance to collect all cases in a database to improve the acknowledgment on fetal ICH.
Thank you for your contribution.
Best regards,
Marta Pallottini